# Bayesian Aggregation Improves Traditional Single-Image Crop Classification Approaches

**DOI:** 10.3390/s22228600

**Published:** 2022-11-08

**Authors:** Ivan Matvienko, Mikhail Gasanov, Anna Petrovskaia, Maxim Kuznetsov, Raghavendra Jana, Maria Pukalchik, Ivan Oseledets

**Affiliations:** 1Yandex LLC, 119021 Moscow, Russia; 2Center for Artificial Intelligence Technology, Skolkovo Institute of Science and Technology, 121205 Moscow, Russia; 3Center for Agro Technologies, Skolkovo Institute of Science and Technology, 121205 Moscow, Russia; 4Public Joint-Stock Company (PJSC) Sberbank of Russia, 117997 Moscow, Russia; 5Marchuk Institute of Numerical Mathematics, Russian Academy of Sciences, 119991 Moscow, Russia

**Keywords:** crop classification, unbalanced classes problem, pixel-wise aggregation

## Abstract

Accurate information about growing crops allows for regulating the internal stocks of agricultural products and drawing strategies for negotiating agricultural commodities on financial markets. Machine learning methods are widely implemented for crop type recognition and classification based on satellite images. However, field classification is complicated by class imbalance and aggregation of pixel-wise into field-wise forecasting. We propose here a Bayesian methodology for the aggregation of classification results. We report the comparison of class balancing techniques. We also report the comparison of classical machine learning methods and the U-Net convolutional neural network for classifying crops using a single satellite image. The best result for single-satellite-image crop classification was achieved with an overall accuracy of 77.4% and a Macro F1-score of 0.66. Bayesian aggregation for field-wise classification improved the result obtained using majority voting aggregation by 1.5%. We demonstrate here that the Bayesian aggregation approach outperforms the majority voting and averaging strategy in overall accuracy for the single-image crop classification task.

## 1. Introduction

Crop identification from satellite images is an important problem in precision agriculture. Possessing accurate information about growing crops makes it possible to regulate agricultural products’ internal stocks and to draw strategies for negotiating agricultural commodities on financial markets. Recent developments in remote sensing and data processing techniques have enabled both scientists and practical workers to simplify the process of crop identification.

The growing use of satellite imagery at very high spatial and temporal resolution allows land managers to obtain extensive data on how the land is used. Different methods for mapping crop patterns by classifying multi-source and multi-temporal data have been proposed and tested by using time series from different satellites in recent years [1]. However, most of these studies demonstrated these methods having certain drawbacks, which were mainly related to the high cost of data collection for time-series and the cloud cover problem, which is commonly faced in different regions. For example, snow cover may stay for more than four months in northern countries, such as Canada or Russia. For some countries, such as India, monsoon season with its dense cloud cover could be the reason for the absence of satellite data. This essentially means that sometimes only one or two cloud-free images can be obtained from satellites and analyzed during the vegetation period. Therefore, an adequate single-image crop classification approach should be used in this case.

Although ML and DL classification algorithms are used to classify crops, the processed classification results’ aggregation toward the field level is not considered in the previous articles. It is also worth noting that most works use multi-temporal data for training. However, how to conduct one-show multi-spectral image learning for crop classification needs to be clarified. The objective of this study was to examine resampling and aggregation techniques for ML and DL models. In this work, we compared ML and DL models, such as convolutions neural networks (CNNs), for crop type recognition. The Sentinel-2 satellite image was used as input data for classification models. We compared classical aggregation methods of pixel-wise classification and also suggested a new approach to aggregate results towards field-wise ones using a Bayesian strategy.

This paper is organized as follows. Section 2 describes the related ML and DL approaches for crop classification, current methods of resampling to tackle class imbalance problems, and recent Bayesian methods applications. Section 3 describes the datasets, satellite image preprocessing and spectral indices generation, ML and DL models used for classification, classical resampling methods, and Bayesian aggregation proposed in this paper. Section 4 presents the results of crop classification, such as overall accuracy and macro-F1 score, and comparison of aggregation methods. Section 5 presents the conclusions.

The main contributions of this paper are as follows:We demonstrated the performance of ML models and the U-Net neural network to address crop classification problems;We compared resampling techniques to tackle the problem of high imbalance of classes;We proposed a new Bayesian aggregation strategy of pixel-wise classification.

## 2. Related Works

Several major approaches are used to tackle the problem of crop classification based on satellite images. For the last decade, classical machine learning (ML) was the most popular and powerful instrument used for these purposes. Many authors have already studied crop type classification using traditional statistical or machine learning methods including Random Forest [2,3,4], Discriminant analysis [5], k-Nearest Neighbors, Extreme Learning Machine [6], Maximum Likelihood Classification [7,8], CART Decision Trees [4,9] and Support Vector Machine (SVM) [7,10,11,12,13,14]. Recently, multi-spectral Sentinel-2 images and SAR Sentinel-1 data were combined for the crop mapping problem supported with the Random Forest model in Germany; for major crops, the F1-score was equal to 0.8 [15].

Class imbalance has a serious effect on crop type recognition. For example, class imbalance explains 40% of the accuracy variability [16]. Recently, class balancing techniques have been compared, showing improved results using SMOTE-based methods [17]. Moreover, researchers have studied the impact of phenology-based methods on the problem of class imbalance, while absence samples for classification were automatically generated using dynamic time warping [18].

A number of researchers reported on the use of neural networks to tackle the problem of crop classification. For example, deep convolutional neural network [19,20,21] and LSTM recurrent neural networks [21] were successfully used for crop recognition. High-resolution Planet satellite tiles and time series of Sentinel-2 tiles were used for the segmentation of crop types in Africa and Germany using 2D U-Net + CLSTM and a 3D-CNN M as reported by Rustowicz et al. [22]. Another example is CNN’s multi-temporal approach with channel-attention, which demonstrates an average F1 score of 0.6 for classifying multi-class crops [23]. However, another paper claims that the one-shot classification based on hyperspectral imaging and deep convolutional networks approach is effective [24]. Spatiotemporal transferability is one of the most challenging problems in crop recognition. Recently, a transferable model based on the U-Net++ architecture was proposed [25].

Statistical methods based on Bayesian theory have become widespread for environmental applications and Earth sciences. An example is the deep Bayesian network applied to develop a robust satellite image classifier to prevent adversarial attacks [26]. Moreover, Bayesian model averaging was used to provide an aggregated probabilistic estimate of soybean crop yield forecast of deep neural networks, such as the 3DCNN (3D Convolutional Neural Network) and ConvLSTM (Convolutional Long Short-Term Memory) [27]. For classifying tree species, Bayesian inference was used, making it possible to achieve an overall classification accuracy of 87% [28]. Recently, Bayesian Deep Image Prior was proposed to downscale soil moisture data from satellite products [29].

## 3. Materials and Methods

### 3.1. Dataset Description

The dataset used in this work was taken from the ‘Farmpins’ Crop Classification Challenge, which was held on the Zindi challenge platform in 2019 [30]. This dataset consists of time series of satellite images and labeled field masks for South Africa dated in 2017. The area of interest is the proximities of the Orange River—the major agricultural region of South Africa.

The satellite image time-series in the dataset include Sentinel-2 scenes. FarmPin provides satellite images of the entire region across 11 time slices per year covering both summer and winter months. Each Sentinel-2 multi-spectral satellite image has 13 bands, and each band represents the intensities of absorbed radiation by the satellite’s sensor in a specific wavelength range. Bands of the Sentinel-2 L1C images have three different spatial resolutions: 60 m, 20 m, and 10 m for the pixel’s side length.

Labeled field mask data represent field masks for each of nine crops: cotton, dates, grass, lucern, maize, pecan, wineyard, intercrop (mixed wineyard and pecan), and the last one is for vacant fields with no plants on them. The organizers of the challenge claimed that all data were verified by them personally and with the help of drones in 2017.

The initial dataset was split into training and testing datasets, which contain 2497 and 1074 fields, respectively. In this work, only the labeled part was used for training and validating the obtained models; therefore, the final number of fields used in this work was 2497. They were further split into the training and validation sets.

The distribution of fields by classes from the dataset is shown in Table 1.

A distinctive feature of the dataset is relatively small areas of the fields. Each field occupies the area of approximately 0.015 km^2^ on average. Hence, fields contain around 150 pixels at 10 m resolution, which is the best available spatial resolution of Sentinel-2 images.

Since pixel-wise classification with each pixel being treated independently was primarily used by us, the number of pixels in the Dates class was not enough to train a model. For this reason, the Dates class was removed from the experiments. Additionally, the Intercrop class was omitted as these fields contain both the Vineyard and Pecan classes.

### 3.2. Satellite Data Preprocessing

The Sentinel 2 L1C satellite images were used as input data for classification. The preprocessing pipeline that has been used in the experiments is described below:1Radiometric calibration.Radiometric calibration is a procedure used to convert meaningless pixel values (DNs) into physical values such as the intensities of light reflected from all reflecting media and absorbed by the satellite filter. Reflected light from the ground and atmosphere is denoted as Top of the Atmosphere (TOA) reflectance. To derive an image from an uncalibrated image, gain and offset should be applied to raw pixel values (DNs). Typically, information about gain and offset is received from the data provider or obtained from the satellite image metadata.2Atmospheric correction.Since light reflected by plants is worked with, atmospheric correction procedure should be applied, which converts TOA reflectance into surface or BOA reflectance (BOA—bottom of the atmosphere). Both of these procedures, radiometric calibration and atmospheric correction, were performed with the help of the Sentinel Application Platform (SNAP) tool [31]. Radiometric correction is automatically applied to uploaded Sentinel images. Atmospheric correction was carried out using the Sen2Cor plug-in, which is freely distributed by Sentinel.3Spatial resampling.As it was mentioned above, Sentinel images contain bands of different resolutions: 60 m, 20 m, and 10 m per pixel’s side length. Bands with the 60 and 20 m resolution were resampled using bilinear interpolation to the resolution of 10 m per pixel’s side length.4Band normalization.Band normalization is a standard procedure used to normalize each band of a satellite image. Pixel values (normalized BOA reflectance in this case) were used as an input or baseline of the model. The following formula was used for the calculations: ${\hat{x}}_{ij}=\frac{x_{ij}-\mu_{j}}{\sigma_{j}}$, where ${\hat{x}}_{ij}$ is the *i*-th pixel in a normalized *j*-th band in the image *x*; $\mu_{j}$ and $\sigma_{j}$ are the mean pixel value of *j*-th band and its standard deviation, respectively. This normalization improves the robustness of the trained models.

### 3.3. Feature Generation

Spectral vegetation indices (SVI) were used as additional features in the classification models. SVI quantifies the contribution of vegetation properties and allows reliable spatial and temporal inter-comparisons of terrestrial photosynthetic activity and canopy structural variations. Since different plants have different spectrum reflectance rates, SVI will also differ from one crop to another [32]. Following is a list of the used vegetation indices:Normalized Difference Vegetation Index (NDVI).NDVI is the most well-known vegetation index in agriculture. NDVI is considered to be an indicator of crop health. It has been shown that there is a strong correlation between NDVI and the amount of green vegetative mass [33]. The NDVI was calculated as follows:
(1)$$\mathrm{NDVI}=\frac{\mathrm{NIR}-\mathrm{Red}}{\mathrm{NIR}+\mathrm{Red}}\;.$$Enchanced Vegetation Index (EVI).EVI is the optimized version of NDVI designed to have a slightly higher sensitivity in the regions with high biomass amounts. In this study, the 2-band version of this index was used [34]. The EVI was calculated as follows:
(2)$$\mathrm{EVI}=2.5\frac{\mathrm{NIR}-\mathrm{Red}}{\mathrm{NIR}+2.4\;\mathrm{Red}+1}\;.$$Normalized Difference Red Edge Index (NDRE).NDRE is similar to NDVI, except that Red Edge (RE) Band (Band 6) is used instead of Red Band. The NDRE was calculated as follows:
(3)$$\mathrm{NDRE}=\frac{\mathrm{NIR}-\mathrm{RE}}{\mathrm{NIR}+\mathrm{RE}}\;.$$Modified Soil-Adjusted Vegetation Index (MSAVI).MSAVI is the index used to minimize bare soil effects on the Soil Adjusted Vegetation Index (SAVI) [35]. The modified soil-adjusted vegetation index (MSAVI) is a vegetation index used for areas with large patches of bare soil. These areas are unsuitable for NDVI due to the small vegetation areas and the absence of chlorophyll. The MSAVI was calculated as follows:
(4)$$\mathrm{MSAVI}=\frac{2\;\mathrm{NIR}+1+\sqrt{{\left(2\;\mathrm{NIR}+1\right)}^{2}-8\left(\mathrm{NIR}-\mathrm{Red}\right)}}{2}\;.$$

These features were calculated from BOA reflectances before the normalization step of the preprocessing pipeline and were added as additional channels. In total, the input tensor for classification models contained 17 channels.

### 3.4. Resampling Algorithms

Classification of crops via satellite data related to class imbalance data, since in most agricultural regions several major crops and many minor crops are grown [16]. The small area of fields in the study region has an additional impact on the imbalance of classes. In order to overcome high imbalances of classes, resampling techniques were applied to the input data. These techniques are as follows:Random Over-Sampling (ROS).Random Over-Sampling consists in randomly sampling new objects from the set of available objects with replacement for each class [36].Random Under-Sampling (RUS).The Random Under-Sampling method is quite similar to ROS. However, instead of sampling new objects, a subset is formed from the existing one by randomly choosing objects with equal probability for every sample [37].Synthetic Minority Over-Sampling Technique (SMOTE).SMOTE [38] is a more advanced technique for dataset resampling, where, in contrast to ROS and RUS, only new samples are created. For a randomly picked sample from the dataset $x_{base}$, a set of k-nearest neighbors is considered. Then, a corresponding sample $x_{corr}$ is chosen from the set of these neighbors’ objects. The new sample is generated according to $x_{new}=x_{base}+\lambda \;\left(x_{corr}-x_{base}\right)$, where λ∈[0,1] and is sampled using uniform distribution.

Resampling techniques to overcome unbalanced classes used with ML models, such as ROS, RUS and SMOTE are not suitable for convolutional neural networks. Instead of resampling techniques, it is a common practice to assign the weights for each class and use them in the loss function [39]. It allows to reduce the effect of majority class gradient direction dominance in the backpropagation process during training.

The weighted loss is calculated according to the formula:(5)$$L_{w}\left(D\right)=\frac{1}{\left|D\right|}\sum_{(x,y)\in D}w_{y}\;L\left(x,y\right)\;,$$
where L is a certain loss function, *D* is a dataset, (x,y) is a sample–label pair and $w_{y}$ is the weight for class *y*.

### 3.5. Data Augmentation

A data augmentation technique was used to artificially increase the dataset by transforming the initial dataset samples to prevent model overfitting and even the imbalances between classes. Augmentation methods were applied for classification tasks using multi-spectral images [40,41]. The Python library Albumentations was used to build the augmentation pipeline [42]. In the framework here, 100 by 100 pixel images we used. The following augmentation pipeline was used:1Pad the image with reflection mode by 50 pixels on each side;2Rotate the image by a random angle in the (−60,60) degree range;3Apply affine transformation with random parameters in the (0.75,1) range;4Shift image by a random number of pixels in the (−20,20) range along both axes;5Crop in the center down to the size of 100 by 100 pixels.

In all image transforms, bilinear interpolation was used; in label transforms, 0-order transformations were used. Both satellite images and labels were transformed using this pipeline. Output augmented images were fed to the neural network after that.

### 3.6. Classical Pixel Aggregation Methods

We used a pixel-wise approach to train the model, with probabilities p(k|x) of a certain pixel being represented by the feature vector *x* to the class *k*. However, the results of field classification may contain more valuable data for farmers and decision-makers than those generated by pixel-based classification; for this reason, we focused on this type of crop classification. As the classification method has a certain degree of misclassification, a number of pixels within fields were assigned to the wrong classes. At the same time, there is prior information that all pixels from a particular field belong to one class; therefore, we decided to remove misclassified pixels by assigning all pixels within one field to a single class. The classical methods applied for this purpose are average voting and majority voting.

#### 3.6.1. Majority Voting

In machine learning, majority voting is a primary type of aggregation, which enables algorithms to make decisions about the best classifications. For our case, it is formulated as follows: for each pixel $x_{i}$, we deduce its prediction $c_{i}$ by taking a class with maximum probability and then choosing for the whole field the class *c* with the maximum number of classified pixels. Here, $x_{i}$—features of the *i*-th pixel of the field, $p(\mathrm{class}=k|\;x_{i})$—probability of the *i*-th pixel being classified to class *k*, and $\#c_{n}$ is the number of pixels classified to class $c_{n}$. Formulae for this rule:(6)$$c_{i}=\arg\max_{k}p\left(\mathrm{class}=k|\;x_{i}\right)\;,$$
(7)$$c=\arg\max_{n}\#c_{n}$$

The main disadvantage of this aggregation rule is that it does not take into account the confidence of prediction for each pixel. In a situation where the majority class has fewer pixels but with a more confident prediction than the minority class, the less confident class will be chosen.

#### 3.6.2. Average Voting

Averaging is another popular machine learning strategy. It takes the mean predicted probability $\bar{p}\left(\mathrm{class}=k\right)$ of each pixel’s class *c* among all crops in a field *F* and uses that as the final class for that field:(8)$$\bar{p}\left(\mathrm{class}=k\right)=\frac{1}{\left|F\right|}\sum_{i\in F}p\left(\mathrm{class}=k|\;x_{i}\right)\;,$$
(9)$$c=\arg\max_{k}\bar{p}\left(\mathrm{class}=k\right)\;.$$

Average voting is a useful technique; however, the result of this aggregation method can be misleading. Usually, each pixel is treated as a new source of information about the whole field, especially when one has a field with quite a few pixels. Each new pixel that predicts some class with positive probability (greater than 0.5) should increase the confidence of the overall prediction of that class. However, that does not necessarily happen when using average aggregation. For example, consecutive pixels with ’positive’ (higher than 0.5) probabilities—0.6,0.7,0.8—will obtain 0.7 as the overall probability of some class (here, the pixel with probability 0.6 canceled out the one with a high probability of 0.8), although they all are ’positive’ and final confidence should be higher than 0.8.

### 3.7. Bayesian Aggregation

The natural way of incorporating prior information about a pixel’s class into a decision is to use Bayes’ theorem. Let F be the set of pixel features from some field: $x_{i}\in F$. Summing up the log-odds of predictions from every pixel (Equation 10), the final decision is obtained using (Equation 11):(10)$$I\left(k\right)=\sum_{x\in F}\log\frac{1-p\left(k|x_{i}\right)}{p\left(k|x_{i}\right)}\;,$$
(11)$$c=\arg\max_{k}\frac{1}{1+\exp(I(k))}\;.$$

Before applying Bayesian aggregation, it may be useful to perform smoothing of raw predictions in the following way:(12)$${\hat{p}}_{\alpha }\left(k|x_{i}\right)=\alpha p\left(k|x_{i}\right)+\frac{1-\alpha }{N-1}\left(1-p\left(k|x_{i}\right)\right)\;,$$
where α∈(0,1) is a smoothing factor and *N* is the number of classes for classification. The choice of factor α depends on the methods used. A grid search was conducted here, which revealed that for N=7 as in the present case, values α∈(0.3,0.4) perform best. Figure 1 presents the process of pixel aggregation.

### 3.8. Crop Classification Pipeline

Using preprocessed bands and generated features, ML models and U-Net were trained and pixels were classified into classes according to the field dataset. First, satellite data were preprocessed and SVIs were computed as additional features. Second, the initial dataset with crop labels was divided into the training and test sets. After that, ML models and U-Net were trained using different resampling techniques and pixel-wise classified results were obtained. The following machine learning and deep learning methods were applied:k-nearest neighbors (kNN) [43];Random Forest (RF) [44,45];Gradient Boosting Decision Trees (GB) [46];U-Net architecture with SE-blocks [47].

The architecture of the U-Net and SE-blocks network is shown in the Figure 2. Finally, pixel aggregation techniques were used to forecast crop types on a field-scale level. To evaluate the performance of the ML pipeline, overall accuracy and Macro F1-score were used [48]. Hyperparameters and other technical details of the applied machine learning models are presented in Appendix A.

## 4. Results and Discussion

In this section, we present the results of our experiments on crop classification, as well as the comparison of various resampling techniques and pixel aggregation methods.

Table 2 shows the main results of pixel-wise crop classification before applying the aggregation procedure. It provides the assessment of ML models and U-Net neural network performance using overall accuracy and F1 score. The table also compares the results of different resampling techniques: ROS, RUS, SMOTE and weighting.

Table 2 demonstrates the differences in the performances between ML models and U-Net. The U-Net neural network with SE-blocks (U-Net+SE) outperformed all other methods, namely, GB, kNN classifier and RF. Its OA was as high as 70.1 and Macro F1 score as high as 0.57. When classical machine learning models were compared, GB showed the best result (OA 68.7, Macro F1 0.50). The comparison of resampling techniques demonstrated that all the techniques used give more or less similar results in terms of the Macro F1 score. However, the choice of resampling technique greatly affected the OA metric. Weighting resampling outperforms the RUS technique by more than 10% in the case of gradient boosting (GB). Generally, Weighting resampling appears to be a better choice than ROS, RUS and SMOTE to tackle the unbalanced classes problem.

Figure 3 presents the result of crop classification using the U-Net neural network. Different classes of crops on the map are marked with different colors. As we can see on the map, there are fields containing almost equal numbers of pixels assigned to different classes. Such cases pose challenges in field-wise classification, as it is unclear how to make a final decision about the crop class. To address this issue, we tested various aggregation techniques.

Table 3 presents the comparison of different aggregation techniques performance towards field-based crop classification. The effect of aggregation on the results is promising: we applied three different aggregation approaches, and all of them led to a marked improvement in crop recognition results at field scale. Bayesian aggregation showed the best OA among the tested methods. The U-Net neural network shows similar OA metrics for both pixel-wise and field-wise crop classifications, probably because U-net considers particular contextual information.

According to our results, RF, kNN and GB performed sufficiently well in the case of crop classification based on a single satellite image. It also appears that ROS and weighting techniques produce the best overall accuracy and Macro F1 score among the tested methods. To the best of our knowledge, this is the first study where the Bayesian aggregation approach was successfully applied to improve crop classification at the field scale.

## 5. Conclusions

We compared the performances of classical machine learning methods and the U-Net neural network for the crop classification task in South Africa and achieved reasonable performance using aggregation strategy. We assessed and compared the performance of different classical aggregation strategies and suggested a new one based on Bayes theorem, which has never been tested in applied tasks. Bayesian aggregation outperformed other aggregation strategies—namely, majority voting by 1.5% and averaging approach by 0.6%. Consequently, crop classification with Bayesian aggregation is a promising approach for industrial use in agriculture. Moreover, it could be used in other geospatial applications for classification goals, such as forest taxation, land use and land types recognition and others.

## Figures and Tables

**Figure 1 sensors-22-08600-f001:**
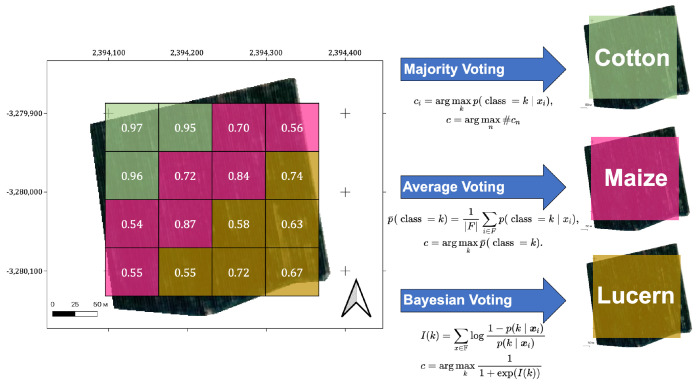
Theoretical visualization of pixel aggregation for the classification model output. Average voting, majority voting and Bayesian voting produce different classes for fields with some pixels assigned to the wrong class.

**Figure 2 sensors-22-08600-f002:**
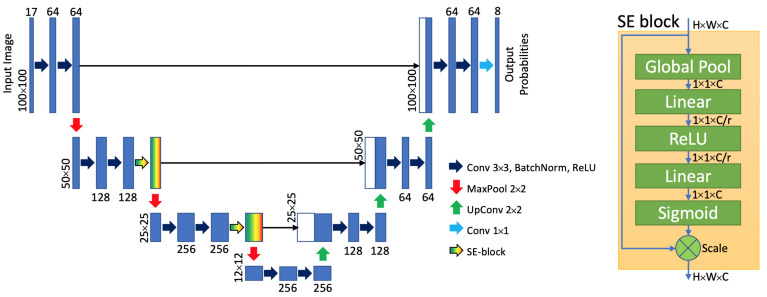
Architecture of U-Net with SE-blocks used for crop classification.

**Figure 3 sensors-22-08600-f003:**
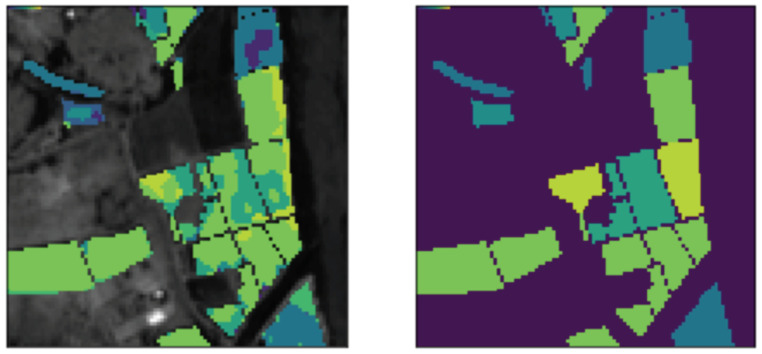
Classification map produced by U-Net+SE setup (**left**) and correct crop labels (**right**).

**Table 1 sensors-22-08600-t001:** Number of fields for each crop after preprocessing of Farmpin dataset.

Cotton	Dates	Grass	Lucern	Maize	Pecan	Vacant	Vineyard	Intercrop
113	2	85	468	251	135	233	789	71

**Table 2 sensors-22-08600-t002:** Comparison of resampling techniques and pixel-wise classification results for ML models and U-net before pixel aggregation. The table does not contain metrics for resampling methods incompatible with some algorithms. For example, ROS, RUS, and SMOTE methods are not applicable for CNN, and weighting is not for kNN.

Classifier	ROS		RUS		SMOTE		Weighting	
	OA	Macro F1	OA	Macro F1	OA	Macro F1	OA	Macro F1
kNN	53.0	0.39	43.5	0.33	53.3	0.40	—	—
RF	62.7	0.46	50.3	0.37	62.4	0.46	62.5	0.46
GB	68.5	0.51	55.3	0.42	65.7	0.50	68.7	0.50
U-Net+SE	—	—	—	—	—	—	70.1	0.57

**Table 3 sensors-22-08600-t003:** Comparison of different aggregation techniques and resampling techniques for field-based crop classification by ML models and U-Net. The table does not contain metrics for resampling methods incompatible with some algorithms. For example, ROS, RUS, and SMOTE methods are not applicable for CNN, and weighting is not for kNN. Bold text indicates the best overall accuracy for each of the resampling methods. For all methods, the Bayesian strategy of pixel aggregation improved overall accuracy.

		Overall Accuracy (%)			
	**Aggregation**	**kNN**	**RF**	**GB**	**U-Net+SE**
ROS	Bayesian	72.66	73.61	77.44	—
	Averaging	72.28	73.42	77.25	—
	Majority	72.08	72.08	76.67	—
RUS	Bayesian	61.41	60.11	70.44	—
	Averaging	61.41	59.92	70.08	—
	Majority	58.62	58.48	69.69	—
SMOTE	Bayesian	72.85	73.23	77.44	—
	Averaging	73.04	73.04	77.44	—
	Majority	72.66	71.70	75.72	—
Weighting	Bayesian	—	72.85	77.44	71.58
	Majority	—	72.47	77.06	71.34
	Averaging	—	71.13	75.91	71.22

## Data Availability

Input data used for research are freely available from the Zindi Platform—https://zindi.africa/competitions/farm-pin-crop-detection-challenge/ (accessed on 1 September 2022).

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
