# Peer review of "Bayesian Aggregation Improves Traditional Single-Image Crop Classification Approaches"

_sensors, 2022, doi:10.3390/s22228600_

Round 1

Reviewer 1 Report

I have major concerns related to the submitted manuscript "Bayesian aggregation improves traditional single image crop classification approaches".

First very high plagiarized manuscript. It is not ethical and shows the clear violation of the journal policy.

Title proposed "Bayesian Aggregation" but the term is not clearly explained or describe in the manuscript, only few lines are not enough, at least include one whole section.

Authors cited most of the old references and also didn't compare their results with other existing studies, that shows the lackness in the novelty of the manuscript.

In data augmentation authors also not cited the original reference used for the enhancement of such type of data.

My recommendation is reject and should be re-submitted after incorporating the following points , more important , avoid plagirism.

Author Response

First of all, we would like to thank the reviewers for their highly relevant and important comments. We carefully consider all of the suggestions and, we believe that we were able to improve the manuscript based on reviewers' recommendations significantly. Our response to the comments is organized in the following way: at first we put the reviewer’s comment, then we provide the answer.

Reviewer point 1: I have major concerns related to the submitted manuscript "Bayesian aggregation improves traditional single image crop classification approaches".

First very high plagiarized manuscript. It is not ethical and shows the clear violation of the journal policy.

Author response: As far as we know, there is no plagiarism in the original manuscript. Automatic plagiarism detection systems may find some level of duplication due to a preprint of the original manuscript - arXiv (https://arxiv.org/pdf/2004.03468). We rewrote duplicate parts and added related works section.

Reviewer point 2: Title proposed "Bayesian Aggregation" but the term is not clearly explained or describe in the manuscript, only few lines are not enough, at least include one whole section.

Author response:  Thank you for your suggestion. We have added a special section devoted to Bayesian aggregation, and we have also described it in more detail.

Reviewer point 3: Authors cited most of the old references and also didn't compare their results with other existing studies, that shows the lackness in the novelty of the manuscript.

Author response:  Thank you for your comment. We agree with it, and have added a related works section to the article. We also added information about various classification satellites, machine learning algorithms, and deep neural networks, as well as the results of using Bayesian methods in other articles.

Reviewer point 4: In data augmentation authors also not cited the original reference used for the enhancement of such type of data.

Author response:  Python library Albumentations were used to build augmentation pipeline. We added citation to this framework and other research devoted to augmentation of multispectral images for classification.

Reviewer 2 Report

This paper investigates the leanring-based image crop classification task. Experiments have been conducted to show the results.

1. The motivations and advantages of Bayesian aggregation should be more explicit.

2. There lack the technical details of resampling methods, which is one of the main contributions of this work.

3. Many numbers are missing in tables 3 and 4.

4. Since the Bayesian theory is essential in the paper, other image processing works based on this are recommended to be included as references, such as A Bayesian deep image prior downsampling approach for high-resolution soil moisture estimation, Deep Bayesian image segmentation for a more robust ejection fraction estimation, etc.

5. It would be better to include state-of-the-arts for experimental comparisons.

Author Response

First of all, we would like to thank the reviewers for their highly relevant and important comments. We carefully consider all of the suggestions and, we believe that we were able to improve the manuscript based on reviewers' recommendations significantly. Our response to the comments is organized in the following way: at first we put the reviewer’s comment, then we provide the answer.

Reviewer point 1: The motivations and advantages of Bayesian aggregation should be more explicit.

Author response: Thank you for your suggestion. We have added a special section devoted to Bayesian aggregation, and we have also described it in more detail.

Reviewer point 2: There lack the technical details of resampling methods, which is one of the main contributions of this work.

Author response:  Thank you for your comments. In Section 3.4 Reasons for Resampling Methods , we have added more information about resampling methods with links to original research, as well as an overview of recent articles on resampling imbalance classes in crop type recognition tasks.

Reviewer point 3:  Many numbers are missing in tables 3 and 4.

Author response:  We added reasons of missing values to each table. The tables don’t contain metrics for resampling methods incompatible with some algorithms. For example, ROS, RUS, and SMOTE methods are not applicable for CNN, and weighting is not for kNN.

Reviewer point 4:  Since the Bayesian theory is essential in the paper, other image processing works based on this are recommended to be included as references, such as A Bayesian deep image prior downsampling approach for high-resolution soil moisture estimation, Deep Bayesian image segmentation for a more robust ejection fraction estimation, etc.

Author response:  Thank you for your suggestion and useful articles. We added a related works section, in which we also reviewed the use of Bayesian methods for classification problems using deep neural networks and other applications in Earth sciences.

Reviewer point 5: It would be better to include state-of-the-arts for experimental comparisons.

Author response:  Thank you for your valuable suggestion, we have added the related works section and described in it state-of-the-art approaches to classifying cultures using machine learning methods and deep neural networks.

Reviewer 3 Report

The paper can be accepted after following changes as:

1. What is the novelty of this work. Highlight it in the revised version of the paper.

2. The abbreviation define once in the start. 

3. Provide organization of paper at the end of introduction section. 

4. How did you find the number of each crop in Table 1?

5. Use comma or full stops at the end of each equation

6. What are the merits of this study.

7. Mentioned the software is used in this study.

8. Use recent references of Bayesian regularisation in this study

Author Response

First of all, we would like to thank the reviewers for their highly relevant and important comments. We carefully consider all of the suggestions and, we believe that we were able to improve the manuscript based on reviewers' recommendations significantly. Our response to the comments is organized in the following way: at first we put the reviewer’s comment, then we provide the answer.

Reviewer point 1: What is the novelty of this work. Highlight it in the revised version of the paper.

Author response: Thank you for yout review and questions. Main novelty of our research in firstly proposed Bayesian strategy method for pixel-wise aggregation of classification results, as well as comparison of one with classical methods of aggregation. To validate proposed method we conducted classification of crops by ML and Dl methods, and compared resampling techniques to tackle class imbalance problem. To clarify novelty we rewrited introduction section and added related works section.

Reviewer point 2: The abbreviation define once in the start.

Author response:  We removed all repeated abbreviations.

Reviewer point 3:  Provide organization of paper at the end of introduction section.

Author response: Thank you for your recommendation, we added organization of paper at the end of introduction section.

Reviewer point 4:  How did you find the number of each crop in Table 1?

Author response: The Zindi platform provided an original dataset in the format of an SHP-file during the FarmPin challenge. The original dataset contains crop names for 2497 fields and areas in polygon format. Numbers in the table indicate the number of fields for each of the nine crop classes.

Reviewer point 5: Use comma or full stops at the end of each equation

Author response:  Thank you for your remark, we added comma or full stop after each equation.

Reviewer point 6: What are the merits of this study.

Author response: The main merit of our study is the proposed new strategy for aggregating pixel-wise classification masks toward field-wise masks. Previous works use ML and DL methods for pixel-wise classification but ignore aggregation toward field scale. We have evaluated the effect of classical aggregation methods on the metrics of ML and DL algorithms, shown their drawbacks, and proposed our method based on the Bayesian strategy.

Reviewer point 7: Mentioned the software is used in this study.

Author response:  We added to Appendix information about used software, packages and versions for each step of computational experiments.

Reviewer point 8: Use recent references of Bayesian regularisation in this study

Author response: Thank you, we agree with your comment. We made a related works section and added information to it about various classification satellites, machine learning algorithms and deep neural networks, as well as the results of using Bayesian optimization in other articles.

Reviewer 4 Report

Major Comments:

1. In the introduction section’s third paragraph Line 35: The discussed literature needs to be more detailed about the used data, algorithm and achieved results. In order to compare your results with them. 

2. It is needed to use more recent literature especially in Line 43 where recent studies is discussed such as:

Asam, S.; Gessner, U.; Almengor González, R.; Wenzl, M.; Kriese, J.; Kuenzer, C. Mapping Crop Types of Germany by Combining Temporal Statistical Metrics of Sentinel-1 and Sentinel-2 Time Series with LPIS Data. Remote Sens. 2022, 14, 2981. https://doi.org/10.3390/rs14132981

3. Related Work section need to be added.

4. No data was provided on the hyperparameters of the used machine learning algorithms like KNN, RF and GB. 

5. The justification of the configuration of the U-Net’s hyperparameters. Was there an optimization algorithm used? 

6. In the conclusion, a comparison between the algorithms was made to show the effectiveness of the proposed algorithm. However, a comparison with other research is needed to show the true significance of your work. 

Minor Comments:

1. In the introduction section third paragraph Line 35: move the discussion of the previous literature to the related work section. 

2. In the introduction section end, there need to be an organizing paragraph of the paper.

Author Response

First of all, we would like to thank the reviewers for their highly relevant and important comments. We carefully consider all of the suggestions and, we believe that we were able to improve the manuscript based on reviewers' recommendations significantly. Our response to the comments is organized in the following way: at first we put the reviewer’s comment, then we provide the answer.

Reviewer point 1: In the introduction section’s third paragraph Line 35: The discussed literature needs to be more detailed about the used data, algorithm and achieved results. In order to compare your results with them.

Author response: Thank you, we agree with your comment. We made a related works section and added information to it about various classification satellites, machine learning algorithms and deep neural networks, as well as the results of using Bayesian methods in other articles.

Reviewer point 2: It is needed to use more recent literature especially in Line 43 where recent studies is discussed such as:

Asam, S.; Gessner, U.; Almengor González, R.; Wenzl, M.; Kriese, J.; Kuenzer, C. Mapping Crop Types of Germany by Combining Temporal Statistical Metrics of Sentinel-1 and Sentinel-2 Time Series with LPIS Data. Remote Sens. 2022, 14, 2981. https://doi.org/10.3390/rs14132981

Author response:  Thank you for your suggestion and this paper information, we have added the this one and another to related works section, where we described state-of-the-art approaches of crop recognition using machine learning methods and deep neural networks.

Reviewer point 3:  Related Work section need to be added.

Author response: Thank you for your valuable suggestion, we have added the related works section and described in it state-of-the-art approaches to classifying cultures using machine learning methods and deep neural networks.

Reviewer point 4:  No data was provided on the hyperparameters of the used machine learning algorithms like KNN, RF and GB.

Author response: We added hyperparameters values and other technical details of the applied machine learning models in Appendix of manuscript.

Reviewer point 5: The justification of the configuration of the U-Net’s hyperparameters. Was there an optimization algorithm used?

Author response:  UNet neural network training settings and hyperparameters were chosen based on our own experience and the experience of other researchers described in the related works section. The used parameters and settings we added to the Appendix. To optimize the neural network, we chose the Adam optimizer.

Reviewer point 6: In the conclusion, a comparison between the algorithms was made to show the effectiveness of the proposed algorithm. However, a comparison with other research is needed to show the true significance of your work.

Author response: We added state-of-the-art metrics for crop types recognition into related works section, as well as works using Bayesian statistics for classification tasks. In the related works section, we have added the methods used and metrics obtained for crop classification by other research groups.

Reviewer point 7: In the introduction section third paragraph Line 35: move the discussion of the previous literature to the related work section.

Author response:  We have moved the literature discussion and added more information about related works, about deep learning for culture classification and about the use of Bayesian approaches for snapshot classification.

Reviewer point 8: In the introduction section end, there need to be an organizing paragraph of the paper.

Author response: Thank you for your recommendation, we added organization of paper at the end of introduction section.

Round 2

Reviewer 1 Report

Can be accepted in present form

Reviewer 2 Report

The authors have addressed my comments.

Reviewer 3 Report

The paper can be accepted in its present form za